Quantitative trait loci mapping and candidate gene analysis of stoma-related traits in wheat (Triticum aestivum L.) glumes

Li Ning 1
Dong Fanfan 1
Liu Tongtong 2
Yang Jinwen 1
Shi Yugang 1
Wang Shuguang 1
Sun Daizhen sdz64@126.com 1
Jing Ruilian 3
1 College of Agronomy, Shanxi Agricultural University , Taigu , China
2 College of Food Science and Engineering, Shanxi Agricultural University , Taigu , China
3 National Key Facility for Crop Gene Resources and Genetic Improvement/Institute of Crop Sciences, Chinese Academy of Agricultural Sciences , Beijing , China
Winkler Robert
Electronic publication date: 2022 Apr 8
Publication date: 2022
Volume: 10
Electronic Location ID: e13262
Received 2022 Jan 13; Accepted 2022 Mar 22
Copyright: ©2022 Li et al.
Copyright year: 2022
Copyright holder: Li et al.
License: This is an open access article distributed under the terms of the Creative Commons Attribution License, which permits unrestricted use, distribution, reproduction and adaptation in any medium and for any purpose provided that it is properly attributed. For attribution, the original author(s), title, publication source (PeerJ) and either DOI or URL of the article must be cited.
License URL: https://creativecommons.org/licenses/by/4.0/

Keywords: Wheat (Triticum aestivum L.), Glume, Stomata, QTL, Candidate gene

Funding: Science & Technology Innovation Foundation of Shanxi Agricultural University 2020BQ30 Outstanding Doctor Funding Award of Shanxi Province SXYBKY2019040 This work was funded by the Science & Technology Innovation Foundation of Shanxi Agricultural University (2020BQ30) and the Outstanding Doctor Funding Award of Shanxi Province (SXYBKY2019040). The funders had no role in study design, data collection and analysis, decision to publish, or preparation of the manuscript.

==============================
The photosynthesis of wheat glumes makes important contributions to the yield. Stomata play a crucial role in regulating photosynthesis and transpiration in plants. However, the genetic base of wheat glume stomata is not fully understood. In this study, stomatal length (SL), stomatal width (SW), stomatal density (SD), potential conductance index (PCI) of stomata, stomatal area (SA), and stomatal relative area (SRA) were measured in different parts of wheat glumes from a doubled haploid (DH) population and their parents. Quantitative trait loci (QTLs) of these traits were anchored on a high-density genetic linkage map of the DH population. A total of 61 QTLs for stoma-related traits were mapped onto 16 chromosomes, and each one accounted for 3.63 to 19.02% of the phenotypic variations. Two QTL hotspots were detected in two marker intervals, AX-109400932∼AX-110985652 and AX-108972184∼AX-108752564, on chromosome 6A. Five possibly candidate genes (TraesCS6A02G105400, TraesCS6A02G106400, TraesCS6A02G115100, TraesCS6A02G115400, and TraesCS6A02G116200) for stoma-related traits of wheat glumes were screened out , according to their predicted expression levels in wheat glumes or spikes. The expression of these genes may be induced by a variety of abiotic stresses. These findings provide insights for cloning and functional characterization of stoma-related candidate genes in wheat glumes.

Introduction

Stomata are the main portals for the exchange of gas and water between plants and the external environment (Li et al., 2017), and they play an extremely important role in the life activities of plants. Plants optimise their photosynthesis and transpiration rates through regulating the aperture, density, and distribution of stomata when they are stressed by biotic or abiotic factors (Doheny-Adams et al., 2012; Franks et al., 2015). In addition to spreading a large number on the leaves, stomata also exist on the epidermis of certain non-foliar organs, such as pods of soybean and oilseed rape, corn bracts, and ears of wheat. (Wang, Wei & Zheng, 2001; Kong et al., 2010).

Previous studies have reported that leaves are the main organs for plant photosynthesis to generate energy, and the photosynthesis of wheat flag leaves has always been regarded as the main source of assimilation during the filling process (Maydup et al., 2010). However, as scientists continue to deepen the research on plant photosynthesis, more and more results showed that the photosynthesis of plant non-foliar organs also plays an important role in the accumulation of carbon assimilates (Sánchez-Díaz et al., 2002; Zhu et al., 2009; Sánchez-Bragado et al., 2014). Compared with leaves, ear organs have unique advantages: for example, the photosynthetic products of wheat ears can be directly transported to the grain, thus avoiding unnecessary energy waste. Wheat ears carry out the C4 metabolic pathway, which can re-fix the CO2 produced by photorespiration (Knoppik, Selinger & Ziegler-Jöns, 1986; Araus et al., 1993). Wheat spikes have stronger drought tolerance, higher osmotic adjustment ability, and water use efficiency (WUE) (Tambussi & Nogues, 2005). Compared with lower organs such as flag leaves, wheat spikes age more slowly (Tambussi et al., 2007). Thus, wheat ear photosynthesis also makes an important contribution to the yield (Araus et al., 1993; Abbad et al., 2004; Zhou et al., 2014).

Glumes are the main photosynthetic organs of the ear and are believed to be an important source of assimilates for kernel filling in wheat (Araus et al., 1993). Glumes can recycle the CO2 respired by developing grains during photosynthesis and have higher ribulose-1,5-bisphosphate carboxylase (RuBPC, EC 4.1.1.39) activity compared with other ear elements (Gebbing & Schnyder, 2001; Aliyev, 2012; Sánchez-Bragado et al., 2014). It has been reported that glumes actively participate in the process of CO2 assimilation during kernel filling (Lopes et al., 2006). In addition, glumes maintain a higher relative water content and WUE under progressive waterlogging and drought stress than flag leaves, contributing significantly to grain filling (Rubén et al., 2018; Wardlaw, 2002). Therefore, compared with other organs, glumes may have a higher ability to resist abiotic stress.

So far, many reports on genetic analysis of stoma-related traits have focused on plant leaves, especially in rice. Teng et al. (2004) detected one QTL that controls the stomatal conductance of rice leaves at the peak tillering stage on chromosome 4. Ishimaru et al. (2001) used a population of crosses between japonica and indica to detect a QTL that controls the stomatal density on the leaf surface and a QTL that is related to the stomatal density on the back of the leaf. Ten QTLs for stomatal density and four QTLs for stomatal size were detected across growth stages and leaf surfaces (adaxial and abaxial) (Laza et al., 2010). In wheat, twenty QTLs for stomatal density and size of leaves were identified under drought stress (Wang et al., 2016). However, genetic analysis of stoma-related traits in wheat glumes is rarely reported.

In this study, stomatal density, length, and width on the top, middle, and base of wheat glumes were measured and the potential conductance index of stomata, stomatal area, and stomatal relative area in different parts of glumes were calculated. A high-density linkage genetic map was used for QTL mapping of stoma-related traits and the candidate genes related were screened. These QTLs and candidate genes will provide insights for studying the molecular mechanism of stomatal development of the wheat ear.

Materials & Methods

Test material

A wheat double haploid (DH) population (Liu et al., 2013), including 150 lines that derived from a cross between Hanxuan 10 and Lumai 14 was used in this study. All the 150 lines and parents were grown at the experimental farm (37°25′N, 112°35′E) of Shanxi Agricultural University in 2018 and 2019. The field experiments were conducted by randomized complete block design (RCBD) with three replicates. Each plot consisted of two rows of 2 m in length, with 0.25 m between rows. Water and fertilizer management during the growth period was complied with the local production practice.

Measurement and calculation of stomata-related traits in glumes

For each DH line and parents, three flowering plants with consistent growth were tagged. The middle spikes of the three plants were quickly placed into a 2-mL centrifuge tube with FAA solution (formalin: acetic acid: 70% alcohol = 1:1:18) three days after anthesis, respectively. The sampling was started at 10 am on the third day after anthesis. The average temperature at the time of sampling was 13 and 17 °C, and the moisture content of topsoil (0∼30 cm) was approximately 15% and 13% (v/v) in 2018 and 2019, respectively. Stomatal density (SD), stomatal length (SL), and stomatal width (SW) on the top, middle, and base of wheat glumes were measured as previously described in Wang et al. (2018).

Calculations of other stoma-related traits were as follows:

Potential conductance index (PCI) = SL2 × SD × 10−4 (Nicholas & Richardson, 2009)

Stomatal area (SA) = π × 1/2 SL × 1/2 SW (Robert, Gretchen & Richard, 2000)

Stomatal relative area (SRA) = SD × SA × 10−4 (Sun et al., 2021)

Data analysis

The relevant t-test and analysis of variance (ANOVA) were carried out by the statistical software package SPSS v.17.0; the frequency distribution map was generated by Excel 2007; and the phenotypic correlation analysis map was created in the R-package corrplot (Wei & Simko, 2013).

QTL mapping

The genetic map of the DH population was constructed by Jing Ruilian’s team at the Institute of Crop Science, Chinese Academy of Agricultural Sciences (Li et al., 2019). The linkage map is 4,082.4 cM in length and contains 1630 SNP and 224 SSR markers, with 2.2 cM per bin on average (Shi et al., 2020). QTL mapping was performed as previously described in Li et al. (2021) using the IciMapping 4.1 software. The LOD score for declaring a QTL was 2.5 for each trait.

Prediction of candidate genes

Candidate genes in associated loci were predicted according to the reference genome sequence of ‘Chinese Spring’ wheat (IWGSC RefSeqv1.1) published by the International Wheat Genome Sequencing Consortium. Gene annotation was carried out by referring to the Ensembl Plants database (https://plants.ensembl.org/index.html). We used a publicly available database, WheatOmics (http://wheatomics.sdau.edu.cn/) (Ma et al., 2021), to obtain the expression profiles of all candidate genes.

Results

Phenotypic variation of glume stoma-related traits in the DH population and parents

We observed regular rows of stomata on the top, middle, and base of glumes in parents (Fig. 1). SD of different parts of glumes in Lumai 14 showed significant or highly significant differences between the two years, while Hanxuan 10 had a significant difference in SD only at the base of glumes (Table 1). For DH lines, except for SD at the base of glumes, the phenotypic values of SD of other parts of glumes in 2019 were significantly lower than those in 2018 (Table 1). These results indicated that SD of glumes is greatly affected by the environment, and this phenomenon is more obvious in Lumai 14 than Hanxuan 10. In general, from the top to the base, SD of glumes gradually became less (Table 1).

Figure 1 Distribution of stomata in different parts of wheat glumes observed in 2018 (A) and 2019 (B).

Table 1 Phenotypic variation of stoma-related traits of wheat glumes in the DH population and parents.

Traitsa	Environments	Parents	Differencec	DH lines	
		Hanxuan10b	Lumai14b		Min	Max	Meanb	SD	Skewness	Kurtosis	CV	
SDt
(No./mm2)	2018	89.79a/A	81.63a/A	8.16∗	60.21	111.44	85.43a/A	10.50	−0.37	0.81	12.29%	
2019	88.15a/A	74.12b/B	14.03∗∗	59.41	106.87	78.99b/B	8.95	−0.91	3.67	11.33%	
SDm
(No./mm2)	2018	80.27a/A	77.16a/A	3.15	52.03	112.20	78.33a/A	10.42	0.28	0.26	13.30%	
2019	82.17a/A	61.49b/B	20.68∗∗	50.24	94.47	73.56b/B	9.14	−0.85	2.11	12.42%	
SDb
(No./mm2)	2018	75.35b/A	74.01a/A	1.34	43.54	105.46	72.38a/A	10.92	0.16	0.34	15.08%	
2019	82.05a/A	63.13b/A	18.92∗∗	40.01	103.90	69.17a/A	11.08	−0.31	1.98	16.02%	
SLt
(µm)	2018	45.60a/A	47.58a/A	−1.98	39.76	50.83	45.71a/A	2.42	−0.17	−0.61	5.29%	
2019	45.08a/A	45.19a/A	−0.11	40.03	50.25	45.67a/A	2.07	−0.16	−0.51	4.53%	
SLm
(µm)	2018	48.42a/A	46.85a/A	1.57	39.19	51.76	45.99a/A	2.32	−0.09	−0.23	5.04%	
2019	44.50b/A	46.75a/A	−2.25∗	40.11	50.97	46.11a/A	2.24	−0.35	−0.13	4.86%	
SLb
(µm)	2018	48.73a/A	46.96a/A	1.77	38.33	51.25	45.22a/A	2.59	−0.11	−0.19	5.73%	
2019	45.15a/A	47.06a/A	−1.91∗	38.63	50.50	44.70a/A	2.40	−0.03	−0.50	5.37%	
SWt
(µm)	2018	27.47a/A	34.55a/A	−7.08**	24.65	36.50	29.21a/A	2.42	1.06	2.23	8.28%	
2019	30.18a/A	28.93b/A	1.25	25.10	33.54	27.65b/B	1.29	1.28	3.08	4.67%	
SWm
(µm)	2018	30.45a/A	30.80a/A	−0.35	25.53	35.54	29.57a/A	2.22	0.86	1.02	7.51%	
2019	29.03a/A	29.55a/A	0.52	24.72	33.00	27.86b/B	1.58	0.53	0.64	5.67%	
SWb
(µm)	2018	30.80a/A	31.53a/A	−0.73	24.83	35.91	29.34a/A	2.48	0.79	1.08	8.45%	
2019	29.45a/A	32.36a/A	−2.91∗	23.82	34.38	27.59b/B	1.88	0.95	1.40	6.81%	
PCIt	2018	19.67a/A	17.91a/A	1.76	9.95	23.49	17.79 a/A	2.20	−0.04	1..01	12.37%	
2019	17.74a/A	14.88a/A	2.86	5.78	22.31	16.45 a/A	1.89	−0.77	2.78	11.49%	
PCIm	2018	17.82a/A	16.77a/A	1.05	9.55	22.93	16.51 a/A	2.01	0.18	1.20	12.17%	
2019	16.31a/A	13.22b/B	3.09∗	5.51	19.93	15.60 a/A	1.96	−0.76	1.97	12.56%	
PCIb	2018	17.79a/A	16.40a/A	1.39	7.63	21.51	14.75 a/A	2.24	0.09	0.62	15.18%	
2019	16.53a/A	13.75a/A	2.78∗∗	4.86	23.28	13.79 a/A	2.30	−0.04	1.58	16.68%	
SAt
(µm2)	2018	1123.29a/A	1032.60a/A	90.69∗	849.26	1397.99	1050.23 a/A	109.82	0.65	0.65	10.46%	
2019	1021.78a/A	986.60a/A	35.18	830.49	1258.00	991.41 a/A	65.37	0.48	1.21	6.59%	
SAm
(µm2)	2018	944.15a/A	979.19a/A	−35.04	861.54	1460.76	1069.71 a/A	102.88	0.50	0.85	9.62%	
2019	996.95a/A	1043.27a/A	−46.32∗∗	818.67	1306.33	1009.14 a/A	83.93	0.44	0.84	9.47%	
SAb
(µm2)	2018	1135.79a/A	1120.20a/A	15.59	799.39	1412.54	1044.39 a/A	119.29	0.61	0.36	8.32%	
2019	975.76a/A	1153.63a/A	−177.87∗∗	784.21	1203.87	968.12 b/B	85.79	0.31	−0.40	8.86%	
SRAt
(%)	2018	9.96a/A	8.31a/A	1.65	4.56	12.16	8.94 a/A	1.20	0.01	0.74	13.42%	
2019	8.89a/A	7.23b/B	1.66	2.77	9.72	7.81 b/B	0.86	−0.96	2.74	11.01%	
SRAm
(%)	2018	9.09a/A	6.82a/A	2.27∗	4.59	12.16	8.36 a/A	1.25	0.44	0.85	14.95%	
2019	8.08a/A	6.30b/A	1.78	2.62	10.24	7.40 a/A	0.97	−0.58	1.75	13.11%	
SRAb
(%)	2018	8.65a/A	8.52a/A	0.13	4.35	12.16	7.54 a/A	1.31	0.48	0.85	17.37%	
2019	7.96a/A	7.17a/A	0.79	2.34	11.79	6.68 b/A	1.17	0.25	1.91	17.51%	
Notes.

a SD, stomatal density; SL, stomatal length; SW, stomatal width; PCI, potential conductance index; SA, stomatal area; SRA, stomatal relative area; t, top of glumes; m, middle of glumes; b, base of glumes.

b Different letters after the values in a column indicate significant differences between two years.

c Asterisks (* and **) indicate significance at p-value < 0.05 and p-value < 0.01, respectively.

Except for the middle of glumes in Hanxuan10, differences in SL of the rest parts of glumes between the two years were not significant in the two parents. Except for the top of glumes of Lumai 14, differences in SW of the rest parts of glumes between the two years were not significant in the two parents. For the DH lines, the difference in SL of glumes was not significant between the two years, but SW in 2019 was highly significantly smaller than that in 2018 (Table 1). These results indicated that SW of glumes is more affected by the environment than SL. For the 150 DH lines, there were genetic differences in the same stoma-related traits among different lines, and this difference varied among different traits. The coefficient of variation of SD in the DH population was greater than 10 in both 2018 and 2019, while the coefficient of variation of both SL and SW was less than 10. This also showed that different stomatal-related traits are affected differently by the environment. In addition, SL and SW did not change significantly from the top to the base of wheat glumes.

Except for the middle of glumes of Lumai 14, differences in stomatal PCI of the rest parts of glumes between the two years were not significant in the two parents. Differences in stomatal PCI of all parts between the two years were not significant for the DH lines neither. Differences in SA of all parts between the two years were not significant in the two parents. Except for the top and middle of glumes of Lumai 14, differences in SRA of the rest parts of glumes between the two years were not significant in the two parents (Table 1).

Stoma-related traits of the DH lines showed continuous transgressive segregation with skewness and kurtosis values close to zero, suggesting normal distribution. All target traits were thus quantitatively controlled by multiple genes and were suitable for QTL mapping (Table 1) (Fig. S1).

Correlation between glume stoma-related traits

For all parts of glumes, SD showed a highly significantly negative correlation with SL in 2018 and 2019; however, the correlation between SD and SW was not significant. There was a significantly positive correlation between SL and SW in 2018, but such correlation was not significant in 2019. In addition, SD showed a highly significant and positive correlation with PCI and SRA in 2018 and 2019. SL showed a significantly positive correlation with SA in both years. SW showed a significantly positive correlation with SA and SRA in both years. In 2018, any two of PCI, SA, and SRA were significantly positively correlated, while the degree of correlation was weakened in 2019 (Fig. 2 Table S1).

Figure 2 Correlation of stoma-related traits in 2018 (A) and 2019 (B).

Red and blue colors indicate significantly positive and negative correlations, respectively, whereas white color indicates no significant correlation. SD, stomatal density; SL, stomatal length; SW, stomatal width; PCI, potential conductance index; SA, stomatal area; SRA, stomatal relative area; t, top of glumes; m, middle of glumes; b, base of glumes.

Figure 3 Distribution of QTLs for stoma-related traits on a high-density linkage map.

To better display the QTLs, the high-density linkage maps show only markers near the QTL intervals. The QTLs with underscore indicate that they were detected in both years.

QTL mapping for traits related to stomata in wheat glumes

A total of 61 QTLs for traits related to stomata of wheat glumes were detected in the two years. The phenotypic variation of these QTLs ranged from 3.63 to 19.02%. The LOD score ranged from 2.51 to 22.12, and the QTLs were distributed on 16 chromosomes including 1A, 1D, 2A, 2B 2D, 3A, 3D, 4B, 5A, 5B, 5D, 6A, 6B, 7A, 7B, and 7D, respectively (Fig. 3 Table 2). A total of nine QTLs were detected in both years.

A total of 10 QTLs corresponding to SD of glumes were detected in the two years. The phenotypic variation of these QTLs ranged from 6.78 to 11.41%. Among these QTLs, QSDt-2A was detected in both years; QSDt-5A and QSDm-5A were detected in the same interval in 2018. Twenty-one QTLs were associated with stomatal size and detected in both years. Among these QTLs, QSLt-2D, QSLm-7A, and QSLb-6A-1 were detected in both years. QSLb-6A-2 and QSWb-6A were detected in the same interval in 2019. A total of 30 QTLs associated with PCI, SA, and SRA were detected in the two years. Among these QTLs, QPCIb-6A-1, QSAt-2D, QSAm-3A, and QSRAb-6A-2 were detected in the two years. Among all QTLs, QSRAb-6A-1 had the largest LOD value (22.12) (Fig. 3 Table 2).

Among all QTLs, two QTL hotspots were found on chromosome 6A. One was in the interval AX-109400932∼AX-110985652, which contained four QTLs related to SL, SD, PCI, and SRA in the two years. The other was in the interval AX-108972184∼AX-108752564, which contained four QTLs related to SW, SD, PCI, and SRA in the two years (Fig. 3 Table 2).

The prediction of candidate genes

For the first QTL hotspot, the physical positions of the two markers AX-109400932 and AX-110985652 were 73571398 and 76990896 bp, respectively. According to the reference genome sequence of ‘Chinese Spring’ wheat (IWGSC RefSeqv1.1), a total of 33 genes were found between the two markers, and gene annotation was carried out by referring to the Ensembl Plants database (Table S2). For the other QTL hotspot, the physical positions of the two markers AX-108972184 and AX-108752564 were 83931623 and 86272494 bp, respectively, covering a total of 24 genes (Table S2). Using WheatOmics, the expression levels of these 57 genes in wheat glumes and spikes were predicted (The International Wheat Genome Sequencing Consortium IWGSC, 2018). The results showed that there were five genes (TraesCS6A01G105400, TraesCS6A01G106400, TraesCS6A01G115100, TraesCS6A01G115400, and TraesCS6A01G116200) with higher expression levels in wheat glumes or spikes (Fig. 4 Table 3).

Table 2 Quantitative trait loci (QTLs) for stoma-related traits of wheat glumes in DH populationa.

Traitb	Locationc	QTLd	Chr	Left marker	Right marker	2018	2019	
						LOD	PVE (%)	Add	LOD	PVE (%)	Add	
SD	Top	QSDt-2A	2A	AX-95631506	AX-94664024	3.57	9.60	2.73	2.63	6.92	2.28	
QSDt-5A	5A	AX-111662464	AX-95683796	2.66	8.10	−2.94				
QSDt-5D	5D	Xgwm205.2	AX-89390905				2.57	6.78	−2.25	
Middle	QSDm-1A	1A	AX-111105973	AX-94402739				2.82	8.89	2.66	
QSDm-5A	5A	AX-111662464	AX-95683796	3.77	8.74	3.37			
QSDm-6B	6B	AX-109288494	AX-94816765				2.56	7.87	−2.47	
QSDm-7B	7B	AX-108729691	AX-94485866	3.02	6.92	−2.97			
Base	QSDb-3A	3A	AX-111635376	AX-110400859	3.44	8.66	−3.37			
QSDb-5A	5A	AX-95630256	Xgwm291				3.84	11.41	3.76	
QSDb-6A	6A	AX-109400932	AX-110985652				7.27	8.98	3.22	
SL	Top	QSLt-1A	1A	AX-111105973	AX-94402739	5.60	11.33	−0.90			
QSLt-1D	1D	AX-109929813	AX-94979481	3.70	7.31	−0.70			
QSLt-2D	2D	AX-109879970	AX-111066402	5.22	11.06	0.76	4.49	9.26	−0.69	
QSLt-3A	3A	AX-95148936	AX-95658831	4.50	8.93	0.79			
QSLt-5A	5A	AX-95152679	AX-94406985	3.36	6.55	−0.66			
QSLt-5D	5D	Xgwm205.2	AX-89390905				2.54	5.45	0.53	
Middle	QSLm-5A	5A	AX-95152679	AX-94406985	2.79	5.07	−0.62			
QSLm-5D	5D	AX-111117089	AX-108782785				2.88	8.68	0.65	
QSLm-7A	7A	AX-95075884	AX-86176614	5.61	10.57	0.96	3.31	6.05	−0.70	
Base	QSLb-5A	5A	AX-108938187	AX-95152679	3.26	9.53	−0.81			
QSLb-5D	5D	AX-111464164	Xgwm205.2				2.94	9.18	0.70	
QSLb-6A-1	6A	AX-109400932	AX-110985652	6.42	9.64	1.17	5.31	8.97	1.01	
QSLb-6A-2	6A	AX-108972184	AX-108752564				6.97	9.59	−1.19	
SW	Top	QSWt-1D	1D	AX-95141814	AX-95661009	2.83	7.89	−0.69			
QSWt-2D	2D	AX-108832290	AX-110274295				3.86	11.31	−0.43	
QSWt-4B	4B	AX-109464953	Xwmc47	2.58	7.40	−0.67			
Middle	QSWm-3D	3D	AX-108735265	AX-108745742				2.99	8.91	−0.47	
Base	QSWb-3A	3A	AX-95659792	AX-111167455				2.84	6.94	−0.52	
QSWb-3D	3D	AX-108735265	AX-108745742	2.63	10.59	−0.69			
QSWb-5A	5A	AX-95659236	AX-109921026				3.32	8.19	−0.57	
QSWb-6A	6A	AX-108972184	AX-108752564	3.47	8.74	−0.77	4.29	9.75	−0.77	
PCI	Middle	QPCIm-1D	1D	AX-110935476	AX-95630666				2.65	3.63	−0.44	
QPCIm-2B	2B	AX-109283083	AX-95101397				4.15	5.82	0.57	
QPCIm-5B	5B	AX-95652462	Xgwm67				11.52	19.02	−1.03	
QPCIm-7B	7B	AX-108768422	AX-108922344				3.72	5.19	−0.53	
QPCIm-7D	7D	AX-89474682	AX-89589386				5.26	7.52	−0.64	
Base	QPCIb-1D	1D	AX-95141814	AX-95661009				2.71	4.56	−0.59	
QPCIb-6A-1	6A	AX-109400932	AX-110985652	18.42	13.64	3.55	21.79	14.06	3.15	
QPCIb-6A-2	6A	AX-108972184	AX-108752564				17.81	8.65	−2.47	
SA	Top	QSAt-1D	1D	AX-94979481	AX-109507293	3.86	6.87	−42.97			
QSAt-2A	2A	Xcwm138.2	AX-110686688	4.12	9.25	−50.09			
QSAt-5A	5A	AX-95202017	AX-110976396	5.14	9.28	−50.40			
QSAt-2D	2D	AX-109879970	AX-111066402	5.24	8.73	31.24	8.40	13.96	−38.91	
QSAt-5D	5D	AX-109464956	AX-109537966				2.92	4.45	−22.28	
Middle	QSAm-1D	1D	AX-94979481	AX-109507293	4.36	10.99	−43.05			
QSAm-3A	3A	AX-95653062	AX-95235020	3.68	9.18	−39.55	3.01	7.42	−35.62	
QSAm-3D	3D	AX-94381228	AX-111161196				3.16	9.18	−31.81	
QSAm-5D	5D	AX-108877411	AX-109725899				2.71	7.88	28.75	
Base	QSAb-5A-1	5A	AX-95659825	AX-111799065				7.45	13.27	−46.15	
QSAb-2A	2A	Xcwm138.2	AX-110686688				6.14	12.89	−44.81	
QSAb-5A-2	5A	AX-111789373	Xwmc340	2.61	7.88	−42.04			
QSAb-5B	5B	Xgwm499	AX-95241032				3.81	6.65	−32.36	
SRA	Top	QSRAt-3A	3A	AX-95659792	AX-111167455	3.20	8.56	−0.35			
QSRAt-5A	5A	AX-111662464	AX-95683796	3.94	11.30	−0.40			
Middle	QSRAm-3A	3A	AX-95659792	AX-111167455	2.66	10.30	−0.35			
QSRAm-5B	5B	Xgwm335	Xgwm540				5.15	12.63	−0.36	
QSRAm-7D	7D	AX-89589386	AX-110969291				3.97	9.51	−0.31	
Base	QSRAb-1A	1A	AX-111764211	AX-111262687	2.51	7.15	−0.35			
QSRAb-3A	3A	AX-111635376	AX-110400859	4.58	13.48	−0.46			
QSRAb-6A-1	6A	AX-109400932	AX-110985652				22.12	10.74	1.68	
QSRAb-6A-2	6A	AX-108972184	AX-108752564	15.31	8.89	−1.39	18.38	6.82	−1.34	
Notes.

a LOD, LOD value of each QTL; PVE, phenotypic variance explained by QTL; Add, a positive sign means increased effect contributed by Hanxuan 10; a negative sign indicates increased effect contributed by Lumai 14.

b SD, stomatal density; SL, stomatal length; SW, stomatal width; PCI, potential conductance index; SA, stomatal area; SRA, stomatal relative area.

c Top, top of glumes; Middle, middle of glumes; Base, base of glumes.

d Underlined QTLs indicate that they were detected in two years.

Then, the expression levels of these five genes under different abiotic stresses were predicted (Oono et al., 2013; Liu et al., 2015; Nazanin et al., 2019). The results showed that the expression of these genes was induced by different abiotic stresses, and the expression patterns of different genes under the same abiotic stress were also different (Fig. 5). For example, the expression of TraesCS6A01G105400 was significantly reduced after being subjected to low phosphorus stress in wheat shoots and roots. The expression levels of TraesCS6A01G115100 and TraesCS6A01G116200 both increased after being subjected to low phosphorus stress in shoots and roots. The expression of TraesCS6A01G106400 was only significantly increased in the root after being subjected to low phosphorus stress. The expression level of TraesCS6A01G105400 was significantly reduced in wheat seedling leaves subjected to drought stress, heat stress, and their combination. The expression levels of TraesCS6A01G106400 and TraesCS6A01G116200 were significantly increased after six hours of drought stress, heat stress, and their combination. Among these five genes, only the expression level of TraesCS6A01G115400 was significantly decreased after being exposed to salt stress (Fig. 5).

Figure 4 The locations of two QTL hotspots on chromosome and the candidate genes contained in the two regions.

(A) LOD value of each traits. The line indicates the position where LOD is equal to 2.5. SD, stomatal density; SL, stomatal length; SW, stomatal width; PCI, potential conductance index; SRA, stomatal relative area; b, base of glumes. (B) Heat maps of expression of candidate genes contained in the two regions in wheat spikelets and glumes. The five genes below the black arrow are candidate genes that were screened out in this study.

Table 3 Candidate genes screened from QTL regions in this study.

Gene ID	Gene annotation	
TraesCS6A02G105400	50S ribosomal protein L3	
TraesCS6A02G106400	Stress-associated endoplasmic reticulum protein 2	
TraesCS6A02G115100	Purple acid phosphatase	
TraesCS6A02G115400	Calcium-dependent lipid-binding (CaLB domain) family	
TraesCS6A02G116200	ATP-dependent RNA helicase	

Figure 5 Expression of candidate genes under various abiotic stresses.

(A) Low phosphorus stress; (B) drought and heat stress; (C) salt stress.

Discussion

Phenotypic correlation of stoma-related traits in wheat glumes

Stomata are the channel for water and gas exchange in the process of wheat photosynthesis and respiration, which indirectly affect the yield of wheat (Berger & Altmann, 2000). Studies have shown that a variety of environmental factors at different growth and developmental stages of plants can affect the formation of stomata, such as water (Stephens & Waugh, 2017), temperature (Qi & Torii, 2018), light (Boccalandro et al., 2009), and CO2 concentration (Hu et al., 2010). Aasamaa, Sober & Rahi (2001) reported that the stomatal length of forest tree species decreased with increasing drought. For some light-loving crops, the formation of stomata can be promoted by increasing the light intensity. In the present study, stomatal density and stomatal width of wheat glumes showed significant differences between the two years, but stomatal length showed no significant differences (Table 1). These results suggest that stomatal length may have higher stability in response to different environmental conditions than stomatal density and width. In addition, we also compared the differences of the same traits in different parts of wheat glumes, and the results showed that from the top to the base of glumes, the stomatal density gradually decreased, but the stomatal length and width did not change significantly. Moreover, four of the nine stable QTLs mapped were associated with stoma-related traits at the base of wheat glumes. It is speculated that this may be because the stomatal properties at the base of the glume are the most stable compared with other parts.

Franks & Beerling (2009) have reported that the negative correlation between stomatal size and stomatal density helps to adjust the plasticity of stomata, thereby regulating the maximal stomatal conductance of wheat. There was a significantly negative correlation between stomatal density and stomatal size in wheat leaves (Wang et al., 2016). In addition, similar phenomena have been observed in other crops. (Ishimaru et al., 2001; Ohsumi et al., 2007). The present results showed that stomatal density was negatively correlated with stomatal length in each part of wheat glumes in the two years. However, there was no significant correlation between stomatal density and stomatal width. Previous studies found that stomatal length and stomatal width in wheat leaves were significantly positively correlated (Wang et al., 2016). The present results showed that stomatal length and stomatal width were significantly positively correlated in 2018 (Fig. 2 Table S1). Therefore, wheat glumes can improve their adaptability to different environmental conditions by coordinating the relationship among stomatal density, stomatal length, and stomatal width.

Pleiotropy of QTLs for stoma-related traits in wheat

Various studies have found that QTLs of closely related traits may be located on the same or nearby positions on the chromosomes (Fracheboud et al., 2002; Tuberosa et al., 2002). In the present study, QSLb-6A-1 corresponding to stomatal length at the base of glumes, QSDb-6A for stomatal density at the base of glumes, QPCIb-6A-1 for stomatal PCI at the base of glumes, and QSRAd-6A-1 for stomatal relative area at the base of glumes were detected in the interval of AX-109400932∼AX-110985652 on chromosome 6A in 2018 and 2019. In the vicinity of this interval, QSWb-6A, QSLb-6A-2, QPCIb-6A-2, and QSRAb-6A-2 were detected in the interval AX-108972184∼AX-108752564 on chromosome 6A in the two years. QSDt-5A, QSDm-5A, and QSRAt-5A were all located in the interval AX-111662464∼AX-95683796. QSDt-5D and QSLt-5D were located in the interval Xgwm205.2∼AX-89390905. QSDm-1A and QSLt-1A were located in the interval AX-111105973∼AX-94402739 (Fig. 3 Table 2).

In addition, compared with previous studies, we found a QTL QSWt-4B in this study was located within the region of QAGsw4B in the previous study (Wang et al., 2018). QSAb-5A-2 was located within the region of QAGsd5A, QMGsd5A-2, and QAGsl5A. (Wang et al., 2018). QSRAm-5B was located close to the region of QPsd5B, and QSDd-5A was located within the region of QSD5A-2 (Wang et al., 2016).

Therefore, the QTLs for above-mentioned stoma-related traits, which were detected in different parts of wheat, different growth periods, and various environments, were significant markers for stoma-related traits in wheat. Furthermore, these findings implied stomatal density and size of wheat leaves and glumes may be controlled by the same or pleiotropic genes. The markers that were localized within a QTL interval associated with stoma-related traits not only validated the QTL but also provided more closely linked markers. These markers will be useful to reveal advanced wheat varieties in wheat breeding programs based on marker-assisted selection approaches.

Prediction of candidate genes related to stomata in wheat glumes

In this study, 57 genes were found in two intervals, AX-109400932∼AX-110985652 (physical range 73571398-76990896 bp) and AX-108972184∼AX-108752564 (physical range 83931623-86272494 bp), on chromosome 6A, and five candidate genes (i.e., TraesCS6A01G105400, TraesCS6A01G106400, TraesCS6A02G115100, TraesCS6A02G115400, and TraesCS6A02G116200) were screened out, according to their expression levels in wheat glumes or spikes (Fig. 4 Table 3).

The expression level of TraesCS6A01G105400 in wheat glumes and spikes was the highest among all candidate genes (Fig. 4), and its functional annotation was 50S ribosomal protein L3 (Table 3). The TraesCS6A01G105400 expression decreased significantly under low phosphorus, drought, and heat stress conditions (Fig. 5). The functional annotation of the candidate gene TraesCS6A01G106400 is stress-associated endoplasmic reticulum (ER) protein (Table 3). The ER plays a crucial role in the maintenance of cellular homeostasis. ER stress is a widely existed stress mechanism to external stimuli in plants and animals. This pathway maintains the ER homeostasis and alleviates stress damage through regulation of a series of gene expressions (Park & Park, 2019). The expression of TraesCS6A01G106400 was significantly increased under salt stress (Fig. 5). The functional annotation of the candidate gene TraesCS6A01G115100 is purple acid phosphatase (PAP) (Table 3), and its expression was significantly increased under low phosphorus stress (Fig. 5). PAPs are members of the metallo-phosphoesterase family identified from a wide range of plants. PAPs have mostly been studied for their potential involvement in phosphorus acquisition and redistribution because of their ability to catalyze the hydrolysis of activated phosphate esters and anhydrides under acidic conditions (Olczak, Morawiecka & Watorek, 2003). Recent studies also showed that PAPs play important roles in modulating plant carbon metabolism, cell wall synthesis, pathogen resistance, etc (Kaida et al., 2009; Sun et al., 2012; Zhang et al., 2014). TraesCS6A01G115400 was specifically expressed in wheat glumes, and its functional annotation is calcium-dependent lipid-binding family protein (Table 3). Ca2+ is a secondary messenger in plants that regulates virtually all aspects of plant development and responses to environmental stimuli. Ca2+ tends to rapidly rise under abiotic stresses (Bartels & Sunkar, 2005). Several proteins have been reported to be activated or translocated in the presence of Ca2+ including Calcium-dependent lipid-binding protein (Hurley & Misra, 2000). The expression of TRAESCS6A01G115400 in roots of wheat seedlings was significantly decreased under salt stress (Fig. 5). The functional annotation of the candidate gene TraesCS6A01G116200 is ATP-dependent RNA helicase (Table 3). ATP-dependent RNA helicase can be found in many organisms, which is involved in the multi-dimensional metabolism of RNA and plays an important role in plant growth and development, especially in abiotic stress response (Kim et al., 2008). The expression of TraesCS6A01G116200 was significantly increased under low phosphorus, drought, and heat stress (Fig. 5).

The above five candidate genes not only had high expression levels in wheat glumes or spikes, but are also induced by a variety of abiotic stresses. Studies have shown that when plants are subjected to abiotic stress, they can change their photosynthetic rate and transpiration rate by adjusting the stomata size, stomata density, and stomata distribution to deal with bad external environments (Doheny-Adams et al., 2012; Franks et al., 2015). Therefore, it will be greatly helpful to analyze whether these five genes affect the formation of wheat stomata through transgenic experiments in future research, and to explore the mechanism of their functions.

Conclusion

In this study, a total of 61 QTLs for traits related to stomata of wheat glumes were identified in the two years, which were distributed across 16 chromosomes and explained 3.63–19.02% of phenotypic variation. Among them, two QTL hotspots were found in 6A, including four and four QTLs, respectively. Subsequently, five candidate genes were screened out, according to their expression levels in wheat glumes or spikes. The expression of these genes could be induced by a variety of abiotic stresses. Our results provide insights for cloning and functional characterization of stoma-related candidate genes in wheat glumes.

Supplemental Information

Supplemental Information 1 Frequency distribution of stoma-related traits of wheat glumes in DH population

SD, stomatal density; SL, stomatal length; SW, stomatal width; PCI, potential conductance index; SA, stomatal area; SRA, stomatal relative area; t, top of glumes; m, middle of glumes; b, base of glumes.

Click here for additional data file.

Supplemental Information 2 Correlation analysis of stomata-related traits in various parts of glumes

SD, stomatal density; SL, stomatal length; SW, stomatal width; PCI, Potential conductance index; SA, Stomatal area; SRA, Stomatal relative area; t, top of glume; m, middle of glume; b, base of glume; * and ** indicate significance at p-value <0.05 and p-value <0.01, respectively. The bottom left and top right of the table are the correlation coefficients of 2018 and 2019 respectively.

Click here for additional data file.

Supplemental Information 3 Genetic information of two QTL hotspots

Click here for additional data file.

Supplemental Information 4 The original value of stoma-related traits in wheat glumes

Click here for additional data file.

We gratefully acknowledge the anonymous reviewers for their constructive comments. We would also like to thank TopEdit for its linguistic assistance during the preparation of this manuscript.

Additional Information and Declarations

Competing Interests

Author Contributions

Data Availability

The authors declare there are no competing interests.

Ning Li conceived and designed the experiments, performed the experiments, analyzed the data, prepared figures and/or tables, authored or reviewed drafts of the paper, and approved the final draft.

Fanfan Dong and Tongtong Liu performed the experiments, analyzed the data, prepared figures and/or tables, and approved the final draft.

Jinwen Yang and Shuguang Wang analyzed the data, prepared figures and/or tables, and approved the final draft.

Yugang Shi analyzed the data, prepared figures and/or tables, authored or reviewed drafts of the paper, and approved the final draft.

Daizhen Sun and Ruilian Jing conceived and designed the experiments, authored or reviewed drafts of the paper, and approved the final draft.

The following information was supplied regarding data availability:

The raw measurements are available in the Supplementary File.

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
