# Peer review of "Quantitative trait loci mapping and candidate gene analysis of stoma-related traits in wheat (Triticum aestivum L.) glumes"

_PeerJ, doi:10.7717/peerj.13262_

## Round 0.1 · original submission · Major Revisions

Besides the technical details (LOD settings in the different experiments), you should specify the environmental conditions, since they are critical for the stomata openness.

Reviewer 1 ·

Basic reporting

English writing is clear, background is enough, references is proper, the structure and figures, tables are good.Figure 3 and 4 may be integrated as one figure.

Experimental design

1. Because that stomata openness is subjected to environmental changes (such as water, N, K fertilizers, etc.), and 150 lines should have genetically different in response to environmental changes. Please add proper discussion. Additionally, please present the data about the environmental condition, such as temperature, soil and leaf water content, irrigation management.
2. Is LOD value setting different for different traits? Why selected the 2.5? Was the mean of performance value of traits across the two years used for QTL location, or only one year for location? Please clarify. How about the effects of environmental factors on QTL locations?

Validity of the findings

The study mapped QTLs for stomatal length, stomatal width, stomatal density, potential conductance index of stomata, stomatal area, and stomatal relative area were measured in different parts of wheat glumes from a doubled haploid (DH) population and their parents, and provide the genetic understanding on wheat stomata.
What difference on trait performance and QTLs among different parts of wheat glumes? Please specify clearly and discuss in the discussion section.
In Result section “Expression analyses of candidate genes”: the study only found candidate genes but surely identified genes, therefore, for me, it is not appropriate to present the analysis based the previous reported data set. If necessary, the related results may be referred to in the discussion. So, the related statement in Abstract should be deleted.

·

Basic reporting

1. The manuscript is easy to understand, though some minor English errors were identified.
2. Research background and references are well organised.
3. Structure of the manuscript is within the standard.

Experimental design

1. Experimental design is reasonable.
2. SInce the manuscript focused on stomata from wheat glumes, the materials and method well suit the research purpose.

Validity of the findings

Yes. The QTL from the base of the glume is significant with a reasonable LOD value, suggesting the high contributions of these QTL to the studied stomatal traits.
Conclusions are well stated, discussed. There is not much QTL studied in stomatal using a large population.

Additional comments

I think this is a well-written manuscript only focusing on stomatal traits. This work requires a large amount of labour work and patience in data collection and analysis, especially when dealing with a DH population. As one of the reviewers, I appreciate the work done by these authors. However, since this work only focused on stomatal traits that involved no research on abiotic stresses, it is hard to say these QTL has a very close link to abiotic stress (as such of drought, salinity). Moreover, I endorse the confirmation of the candidate genes using the public database. However, it might be better if an RT-PCR could be also performed using representative lines to test the candidate genes.

---

## Round 0.2 · accepted · Accept

Thank you very much for addressing the reviewers' comments.

·

Basic reporting

Since this is the revised version based on the comments from the reviewers, I have no further comments.

Experimental design

Since this is the revised version based on the comments from the reviewers, I have no further comments.

Validity of the findings

Since this is the revised version based on the comments from the reviewers, I have no further comments.

Additional comments

Since this is the revised version based on the comments from the reviewers, I have no further comments.